# High-speed and energy-efficient non-volatile silicon photonic memory based on heterogeneously integrated memresonator

Bassem Tossoun [1] ✉, Di Liang [1,2], Stanley Cheung [1], Zhuoran Fang[1], Xia Sheng[1], John Paul Strachan [1,3] & Raymond G. Beausoleil[1]

Recently, interest in programmable photonics integrated circuits has grown as a potential hardware framework for deep neural networks, quantum computing, and field programmable arrays (FPGAs). However, these circuits are constrained by the limited tuning speed and large power consumption of the phase shifters used. In this paper, we introduce the memresonator, a metal-oxide memristor heterogeneously integrated with a microring resonator, as a non-volatile silicon photonic phase shifter. These devices are capable of retention times of 12 hours, switching voltages lower than 5 V, and an endurance of 1000 switching cycles. Also, these memresonators have been switched using 300 ps long voltage pulses with a record low switching energy of 0.15 pJ. Furthermore, these memresonators are fabricated on a heterogeneous III-V-on-Si platform capable of integrating a rich family of active and passive optoelectronic devices directly on-chip to enable in-memory photonic computing and further advance the scalability of integrated photonic processors.

Over recent years, the demand for high-performance computers (HPCs) capable of efficiently running artificial intelligence applications has grown dramatically. The number of programs which use deep learning training has doubled every 3.5 months, which is much faster than the rate of performance doubling predicted by Moore's law[1]. In addition, learning algorithms are required to be executed in real-time on a massive amount of data produced by the plethora of interconnected smart devices within the Internet of Things (IoT) and edge computing.

Today, AI algorithms utilized by applications such as autonomous driving vehicles and Amazon's Alexa, are implemented using neural networks (NNs), a model inspired by the neuro-synaptic network within the human brain, which is the most energy-efficient computer-to-human knowledge (able to process 10 petaflops of data with only 20 W of power)[2]. The most commonly used hardware for running NNs includes application-specific integrated circuits (ASICs), graphics processing units (GPUs), and field-programmable gate arrays (FPGAs). Current state-of-the-art electronic accelerators consume about 0.5 pJ in processing a single multiply-accumulate (MAC) operation, the most fundamental neural network calculation[3].

While conventional microelectronic processor performance progressed in line with Moore's Law as transistor density increased and multi-core processors developed, they are still fundamentally limited in both speed and power. Joule heating and the charging of metal wires involved in the movement of data constrain the operating speed and dominate the power consumption within electronic neural network hardware[4]. To exacerbate this issue even further, the von Neumann bottleneck and the "memory wall" restrict the bandwidth of data communications between the processor and the memory. Furthermore, digital processing units are also bottlenecked by the clock rate of the processor when computing multiply-accumulate (MAC) operations, the most fundamental neural network calculation[5].

Fortunately, several technological breakthroughs over the last few decades have opened novel opportunities to battle these challenges. Silicon photonics offers a promising solution to dramatically improve the bandwidth and energy efficiency of interconnects for data communications applications including data centers and HPCs[6]. Most recently, silicon photonics has not only been used for data communications, but for non-von Neumann accelerators used for applications

[1]Hewlett Packard Labs, Hewlett Packard Enterprise, Santa Barbara, CA, USA. [2]Present address: University of Michigan, Department of Electrical and Computer Engineering, Ann Arbor, MI, USA. [3]Present address: PGI-14, Forschungszentrum Jülich GmbH, Aachen, Germany. ✉e-mail: bassem.tossoun@hpe.com

such as deep learning[7–12]. Some of the inherent properties of photonics make it a suitable platform for neuromorphic computing such as its high bandwidth of data transmission and parallel operation enabled by unique multiplexing schemes like wavelength division multiplexing (WDM). Furthermore, the processing time scale of a photonic neuron is within picoseconds, which is orders of magnitude higher than that of its electronic counterparts[13].

Because running a task on a deep neural network often can take a significant amount of time, there is a significant benefit to having nonvolatile memory on-chip as it eliminates the static power consumption in holding weight values throughout an inference task. On-chip memory also prevents the need to retrieve results stored on a separate memory chip in between epochs or training steps. In addition, nonvolatile photonic memory is not only useful for data storage but also as part of the computational algorithms running on photonic neuromorphic computers[4]. More specifically, high-speed and low-power nonvolatile photonic phase shifters are essential in enabling a larger variety of machine-learning methods to be executed on integrated optical neural networks. For example, deep neural networks utilizing online training with algorithms such as backpropagation require synaptic weights to be updated frequently. These on-the-fly learning algorithms are scalable, memory-efficient, and can even be used to circumvent the losses compounded by the device imperfections within photonic neural networks as they scale in size and complexity[14,15].

One viable solution to supplying a fast, low-power, nonvolatile memory is the memristor (also commonly referred to as resistive random-access memory or RRAM) which was theoretically proposed by Leon Chua and experimentally demonstrated by HP Labs[16,17]. Memristors (also commonly referred to as resistive random-access memory or RRAM) have proven to be excellent nonvolatile electronic memory devices with high switching speed (~100 ps), low energy switching (~100 fJ), endurance ($10^6–10^8$ cycles), and high density[5,18–22].

In this work, we integrated metal−oxide-based memristive devices within III−V/Si microring resonators to produce memresonators, an

energy-efficient analogue nonvolatile memory on a highly scalable and versatile heterogeneous silicon photonic platform well-suited for integrated photonic information processing circuits. By changing the resistance state of the memristor, we can subsequently tune the optical phase within the waveguide and alter the resonant wavelength of the device. Analogue device operation was shown through the measurement of multiple optical states. Performance records, including retention times of 12 h, an endurance of 1000 switching cycles, switching times as low as 300 ps for SET and 900 ps for RESET, and switching energies of 0.15 pJ for SET and 0.36 pJ for RESET, are demonstrated.

By integrating these memristors on the same chip as photonic neural networks, for example, significant amounts of energy and latency can be saved by avoiding energy lost in the transfer of data from the processor to an external memory chip. Moreover, using these memresonators, weights within photonic neural networks can be stored and updated at high speeds and low energy, enabling the use of the back-propagation algorithm and the ability to train the network on-chip. Finally, this III−V-on-silicon photonic memristive device is based on the same technology developed for a fully active (including optical gain) and passive integrated photonic platform on silicon for large-bandwidth, energy-efficient optical interconnect applications[23]. In fact, the first generation of a heterogeneous III−V-on-silicon technology has been successfully commercialized by Intel in their 300 mm CMOS production line to enable on-chip lasers for over 2 million optical transceiver units each year[24,25].

## Results
### Device design and fabrication
As schematically shown in Fig. 1a, heterogeneous III−V/Si microring resonators (MRRs) of varying radii between 10 μm and 25 μm were fabricated on a silicon-on-insulator (SOI) substrate with a 2 μm-thick buried oxide layer and a 300 nm-thick top silicon layer. GaAs epitaxial device layers are transferred to a 100 mm Si-on-insulator (SOI) substrate by an $O_2$ plasma-assisted direct wafer bonding process[26]. About

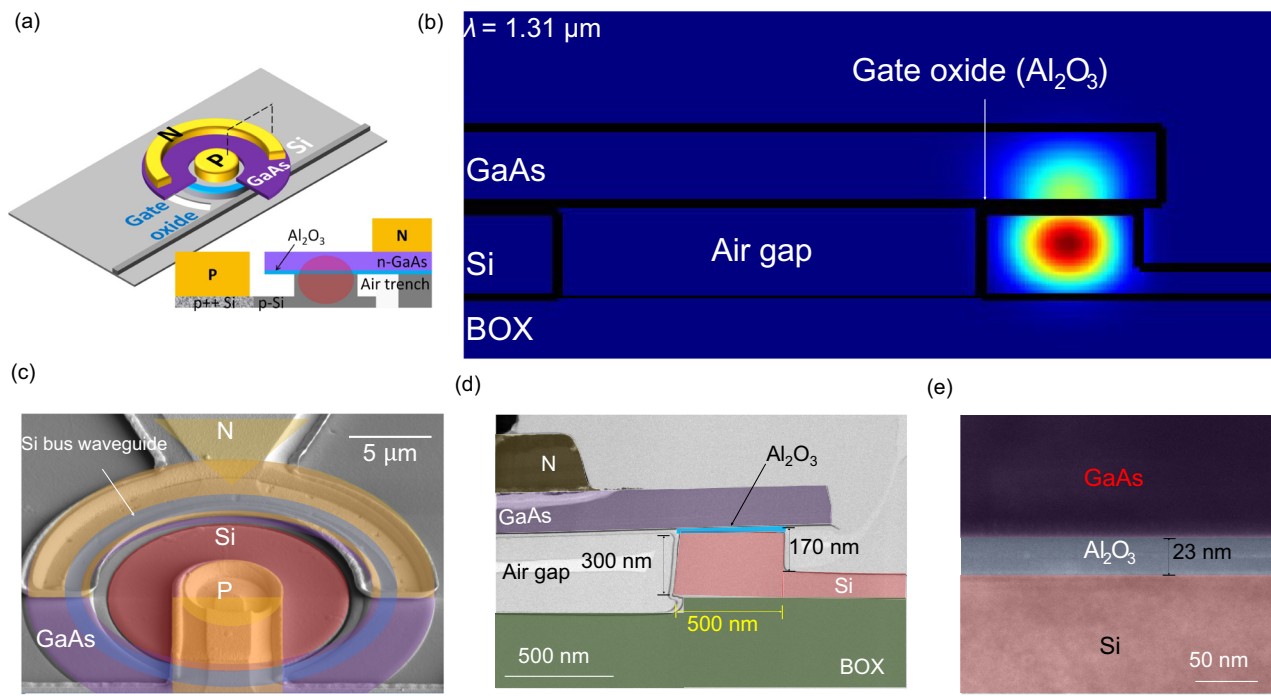

**Fig. 1 | Device schematics and images. a** 3D-view and cross-schematic view of a memristive III−V-on-Silicon microring resonator. **b** Simulated fundamental TE mode field intensity within the microring waveguide at 1310 nm. **c** Scanning electron microscopy (SEM) cross-sectional image of memresonator. **d** Transmission electron microscopy (TEM) cross-sectional image of memresonator. **e** TEM image of a bonded GaAs-Al₂O₃-Si memristor.

10 nm of $Al_2O_3$ was grown on both the GaAs and Si substrates using atomic layer deposition (ALD) before they were bonded together to form the resistive-switching oxide. A n-GaAs/$Al_2O_3$/p-Si semiconductor-insulator-semiconductor (SIS) stack is embedded within the microring resonator for high-speed optical signal modulation through carrier accumulation and the plasma dispersion effect[27]. This device can then be resistively switched like a memristor, thereby producing a memresonator, or a memristor integrated with a microring resonator, as will be discussed in further detail in the next section.

An air trench is then formed on the Si device layer with a ~170 nm waveguide rib etch depth, prior to wafer bonding in order to confine the memristor device area only to the fundamental TE mode and to minimize the area for high-speed and energy-efficient charging and discharging. The bus and ring waveguides within the microring resonators are 500 nm wide each and are separated by 200 nm at the coupling section. Figure 1b is an optical simulation showing the fundamental TE mode within the memresonator waveguide. Transmission electronic microscopy (TEM) images of the fully fabricated memresonator cross section and memristor material stack are shown in Fig. 1d, e, respectively. As seen in Fig. 1a, electrodes are placed on the 150-nm-thick n-type GaAs contact layer and the 300 nm p-type Si contact layer to apply an electrical field across the oxide material. Since semiconductor materials are sandwiching the resistive-switching oxide, these memristors can be integrated within optical waveguides while adding only about 0.05 dB of insertion loss (see Supplementary Note 2), achieving much lower optical loss than with purely metal electrodes typically used in electronic memristors.

## Working mechanism

As mentioned in the previous section, a memristor is formed using n-type GaAs and p-type Si sandwiching a thin resistive-switching $Al_2O_3$ layer. In order to resistively switch the memristor, a process creating an interchange of oxygen and semiconductor atoms, called "electroforming," must be induced by applying a high enough positive bias voltage across the memristor. The high electric field breaks some of the Al−O bonds causing oxygen atoms to migrate towards the semiconductor regions and leave behind negatively ionized vacancies within the $Al_2O_3$ layer. The oxygen vacancies form localized aluminum-rich channels, namely conductive filaments (CFs), that allow current to flow and effectively increase the conductivity of the oxide material, setting the device to a low resistance state (LRS)[28]. When a large enough electric field is applied in the opposite direction, it causes a reduction of oxygen vacancies as well as sufficient current flow to catalyze localized Joule heating, rupturing the CFs previously formed and resetting the memristor back to a high resistance state (HRS)[29,30]. Prior studies suggest that a combination of electric field and Joule heating induces the resistive-switching mechanism of $Al_2O_3$-based memristors[31]. When a positive bias (typically lower than the voltage needed for electroforming) is applied again, the CF reforms, and the device switches back to a lower resistance.

As can be seen in Fig. 2a, a schematic of the resistive-switching mechanism within the memristor is shown. Oxygen vacancies are formed after electroforming, and they can be ruptured and reconnected through subsequent set and reset cycles. Figure 2b, c visually portrays the carrier dynamics within the III–V/Si memristor-integrated waveguide when the memristor is in the LRS and the HRS. Figure 2d shows the current–voltage characteristics of the device which shows a hysteresis-type curve confirming its operation as a memristor. The voltage was swept from 0 to 10 V and back down to 0 V, and then from 0 V to −5 V and back to 0 V to observe the hysteresis effect in the I−V characteristics. The compliance current, $I_{CC}$, was initially set to 50 µA in the forward direction and 1 mA in the reverse direction in order to prevent the device from permanent breakdown and physical damage. Typically, the device had less than 10 nA of DC leakage current in the HRS state due to the high quality of the $Al_2O_3$ (Supplementary Note 3).

The leakage current is mostly due to trap-assisted tunneling through deep-level traps within the $Al_2O_3$ layer. The electroforming step in the memristor occurs at 9 V, the set voltage occurs at around 5 V, and the reset voltage occurs around −4 V.

As can be seen in the current–voltage characteristics in Fig. 2d, the device can also be switched to an intermediate resistance state (IRS) with a resistance between the LRS and HRS by adjusting the current compliance of the measurement equipment to a value between the compliance used for the HRS and LRS. The device can also be set to multiple intermediate in this way, displaying the possibility of using these devices for analogue computing. For example, while the device is in the HRS, it can be switched to the IRS by applying a current compliance and can be switched to a LRS by applying a higher current compliance. Since a lower current compliance is applied, it physically limits the growth of the conductive filament, thereby also limiting the device resistance[32]. Moreover, when the memresonator is set to a low or intermediate resistance state, we found that the conduction in the memristor is observed to be diode-like, which resembles the ideal diode equation, $I \propto [e^{qV} - 1]$. Since the resistive-switching oxide acts as a non-degenerate semiconductor material, and each semiconductor contact layer is p- and n-doped, the device essentially acts like a p–i–n diode in which excess electrons flow from the n-type GaAs to the p-type Si and excess holes flow in the opposite direction[33]. The device begins behaving similarly to a carrier injection type modulator in which majority carriers are injected into the CF and drift from one contact region to the other through the CF. In Fig. 2c, a schematic diagram of this process is shown, displaying electrons being injected from the n-GaAs to the p-Si and holes being injected from the p-Si to the n-GaAs through the CF while the memristor is in the LRS.

## Device characteristics

As shown in Fig. 2e, switching the memresonator between the LRS, IRS, and the HRS subsequently switches its resonance wavelength. The insertion loss was measured to be about 0.047 dB in the HRS and 0.048 dB in the LRS (Supplementary Note 2). The 20-µm diameter memresonator achieves about a 0.08 nm or about a 0.18π phase shift (see Supplementary Note 5) in the LRS, leading to an estimated $V_\pi L$ of 2 V × 0.35 mm = 0.7 mm. The effective refractive index and phase shift as a function of voltage is plotted in Supplementary Fig. 4. After setting the memresonator to the IRS or LRS, the device resonates at the same wavelength until it is reset back to the HRS. In Fig. 3b, e, the resistance in the HRS, IRS, and LRS and the optical power being transmitted through the memresonator at $\lambda_{HRS}$, $\lambda_{IRS}$, and $\lambda_{LRS}$ was measured for 12 h. This measurement demonstrates the non-volatility of this optoelectronic memory device. Drifting in the temperature stability of the setup was observed, which can be mitigated using a temperature-controlled stage. The device also demonstrated repeatability and excellent endurance as it was cycled 1000 times between states using voltage pulses (Fig. 3c). Figure 3f shows the resistance of the HRS, IRS, and LRS, and demonstrates a stable HRS/LRS resistance ratio of about $10^3$ through 1000 switching cycles.

To test the switching speed and energy of these devices, an arbitrary waveform generator was used to generate voltage pulses used for reading and writing the memresonator (see "Methods" for experiment details). The output optical power at the resonant wavelength of the memresonator was monitored as the device was being switched (Fig. 4b, c). These measurements demonstrate the ability to quickly write and read data from the device with ultralow energy. The switching energy is as low as 0.15 pJ, which is more than 30× smaller than the record switching energy for photonic nonvolatile memory devices[34]. After the device was SET using a write voltage pulse, a read voltage pulse was applied to read the optical power transmitted through the memresonator at the resonant wavelength as well as the read current of the memristor, which was 2.5 µA. The normalized transmitted power after the device was SET was about 0.27. The energy

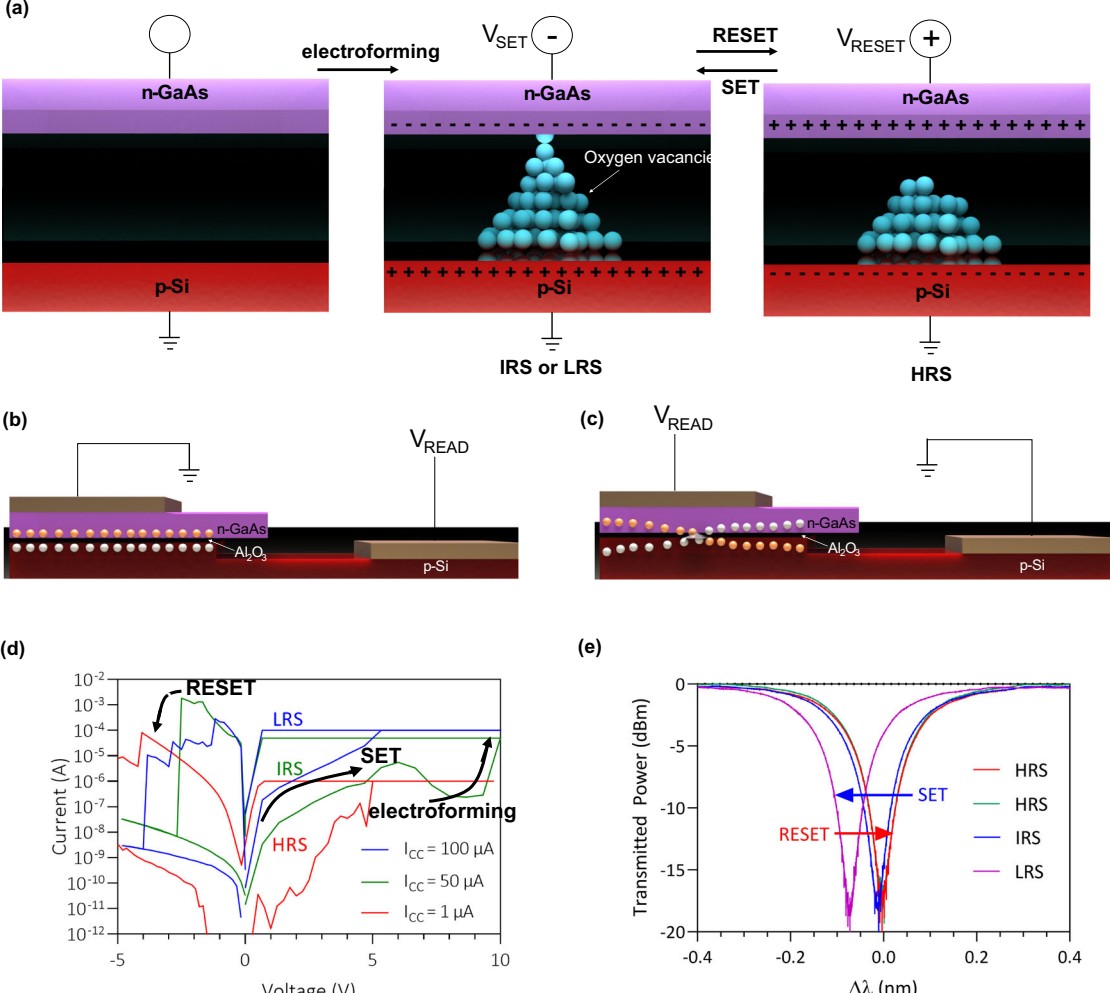

**Fig. 2 | Illustration of device working mechanism and fundamental device characteristics. a** Schematic diagram of the process of forming and rupturing conductive filaments (CFs) within the memristor. $V_{SET}$ is the voltage applied to set the memristor to the IRS or LRS. $V_{RESET}$ is the voltage applied to reset the memristor to the HRS. **b** Schematic diagram of the carrier distribution within the waveguide while a read voltage, $V_{READ}$, is applied to the memristor in the HRS. **c** Schematic diagram of the carrier distribution within the waveguide while a read voltage is applied to the memristor in the IRS or LRS. **d** Current–voltage characteristic of the device displaying the hysteresis signature of a memristor. Current compliances of 1 μA, 50 μA, and 100 μA were used in the forward bias voltage direction in order to set the device into different resistance states. **e** Optical spectrum of the memresonator while a 2 V read voltage is applied in different states.

consumed to read the memresonator after the write cycle was about 5 fJ.

Afterward, the device was RESET using an erase pulse, with a switching energy of 0.36 pJ. Then a read voltage pulse was applied to read the transmission power of the memresonator and the read current of the memristor, which was around 1 nA. The normalized optical power transmitted through the memresonator at the resonant wavelength after the device was SET was about 0.1. There is a small blueshift in the resonant wavelength in the HRS when the read voltage is applied, explaining why there is a small amount of power being transmitted even after the device has been RESET. However, the transmitted read power is nearly 3× times smaller than when the device has been SET. Also, the energy consumption of reading the memresonator after the erase cycle was about 2 aJ. Most importantly, zero static power is consumed in between read and write cycles as energy is only spent during the read and write operations.

The measured switching speed of these devices is over two orders of magnitude faster than the fastest nonvolatile photonic phase shifters and is comparable to all-electronic metal–insulator–metal (MIM) memristor devices[35,36]. Furthermore, electronic memristors made from a similar material stack (Si/SiO₂/Si) have previously shown a SET speed of 7.6 μs and a RESET speed of 490 μs[33]. Ultimately, the switching speed

is limited by a few factors: the atomistic processes within the memristor stack, the time constant associated with Joule heating causing the conductive filament to rupture, and parasitic capacitances[37].

## Discussion

Resistive-switching elements such as memristors have been used for analogue computing for several years. While these electronic memristors can be integrated at high densities within crossbar arrays and switched at high speeds, there exists a trade-off between bandwidth and the total size of the crossbar array. For example, the bandwidth scales inverse proportionally to the size of the crossbar array, meaning that as the size reaches greater than 1 mm², the bandwidth becomes constrained, and the energy cost for off-chip communications can also become problematic. Whereas on a photonic platform, signals can be supported with much greater bandwidth and consume less energy for longer distances than the electrical counterparts. For instance, optical waveguides can be designed with low signal attenuation (< 0.1 dB/cm) and are able to propagate high-power signals without the issues of thermal runaway such as that seen in the Joule heating of electrical wires[38]. Hence, microring-based weight banks and crossbar arrays which perform matrix-vector multiplication operations used for photonic neural networks, as well as for optical content-addressable

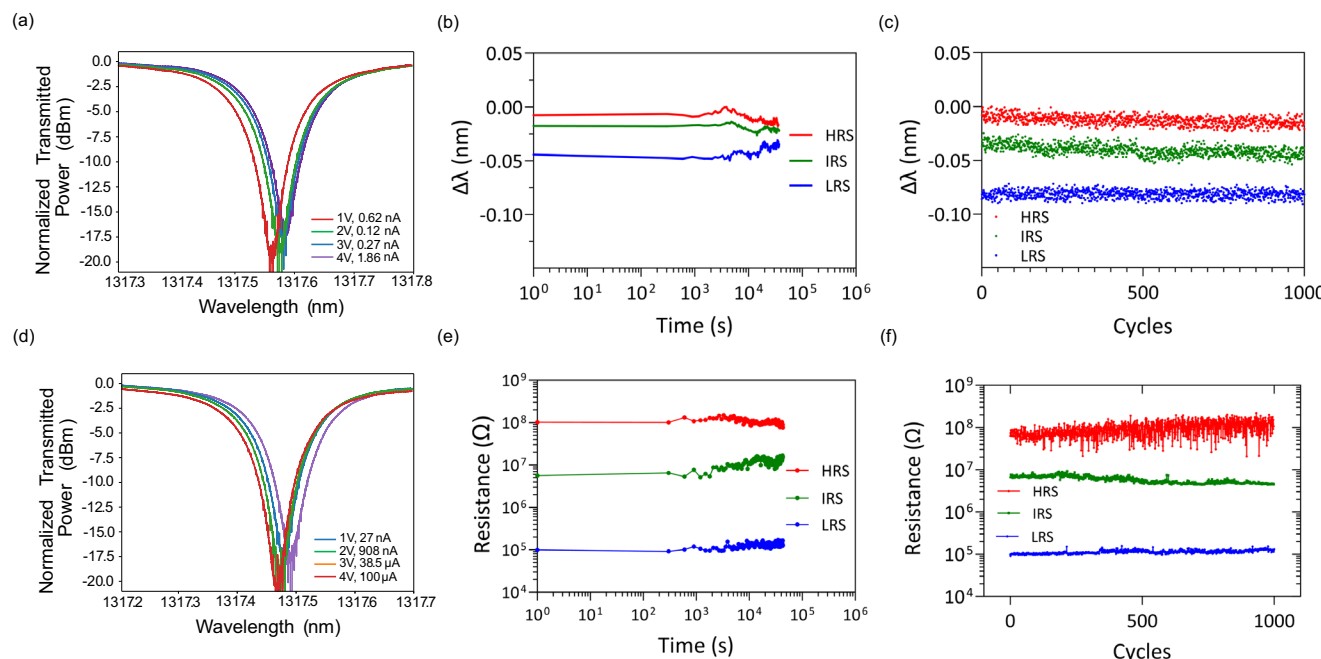

**Fig. 3 | Optical and electrical characteristics of memresonator. a** Normalized transmitted power through the memresonator in the HRS and read current as a function of the read voltage. **b** Wavelength shift of the memresonator in multiple states measured every 5 min over the span of 12 h with a read voltage of 2 V. **c** Wavelength shift of the memresonator in multiple after 1000 set/reset cycles with a read voltage of 3 V. **d** Normalized transmitted power through the memresonator in the HRS and read current as a function of read voltage. **e** Resistance measurements of the memristor in multiple states monitored for 12 h. **f** Resistance measurements of the memristor in multiple states.

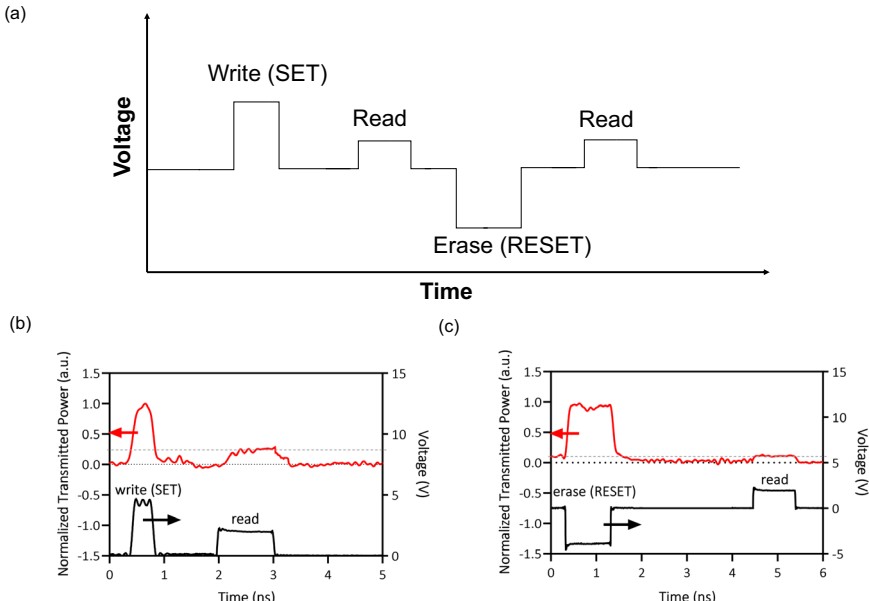

**Fig. 4 | Switching scheme and measured switching high-speed temporal response of memresonator. a** The typical voltage pulse sequence used to write and erase data from a memristor. **b** Plotted is the normalized transmitted power (left *y* axis) at the resonant wavelength of the memresonator as a function of time during a voltage pulse sequence. The voltage of the input pulse sequence is measured on the right *y* axis. The write sequence includes a 5 V amplitude, 300 ps voltage pulse used to SET the device, and a 2 V, 1 ns voltage pulse used to read. **c** The erase sequence includes a −4 V amplitude, 900 ps voltage pulse used to RESET the device, and a 2 V, 1 ns voltage pulse used to read.

memory, can be fashioned out of this platform using these nonvolatile photonic phase shifters[9,39,40]. Furthermore, these types of circuits can potentially achieve larger scales with much higher efficiency than their electronic counterparts.

Table 1 compares the characteristics for different implementations of phase shifters on silicon photonic platforms used for neural networks and optical FPGAs. Typically, thermo-optic phase shifters are used as weights within photonic neural networks, however, since they lack high-speed programming capabilities and nonvolatile memory capabilities, they waste tens of mWs per unit of static power over the span of an inference task and with each weight update cycle. They can also easily cause thermal crosstalk which limits integration density and

**Table 1 | Implementations of programmable phase shifters on a silicon photonic platform**

|  | Thermo- optic[41,42,54–56] | Charge- trapping[57,58] | MEMS[43,59] | PCM[11,34,47,48,60–63] | BaTiO₃ (BTO)[35,50] | Memresonator (this work) |
|---|---|---|---|---|---|---|
| Switching speed | 2.4 μs | > 350 ms | ~1 μs | <100 ns | <1 ms | <1 ns |
| Switching energy | 30.5 nJ | 11.4–17.2 pJ | 0.2 nJ | 180 pJ–17 nJ | 4.6–26.7 pJ | 0.15–0.36 pJ |
| Retention time | N/A | 10 years | N/A | 10 years | 10 h | 12 h |
| $L_\pi$* | ~10 μm | 865 μm | ~1 cm | 11 μm | 1 mm | ~350 μm |
| Insertion loss | 0.23 dB | ~1 dB | 3.5 dB | ~0.3 dB | ~0.1 dB | 0.27 dB** |
| Non-volatility | No | Yes | No | Yes | Yes | Yes |

*$L_\pi$ refers to the length of the phase shifter required to achieve a π phase shift.
**The total insertion loss is measured to be 4 dB at the operating point of interest when including coupling losses.

scale, and control complexity[8,41,42]. More recent demonstrations showed integrated nano-opto-electro-mechanical phase shifters with improved energy efficiency, but were still limited in write speeds (~1 μs), require large switching voltages (> 20 V), and have high mechanical failure rates[43–45].

On the other hand, phase-change materials (PCM) such as Ge₂Sb₂Te₅ (GST) and, more recently, Sb₂Se₃ have been explored extensively as a candidate for nonvolatile memory within silicon PICs with encouraging results[34,46–49]. Recently, they have shown the ability to achieve a π phase shift with less than 12 μm of length and reasonably low insertion losses (< 0.3 dB). However, these materials are also limited in writing speed and typically require high input powers (~mW) to heat them long enough to change the phase from amorphous to crystalline. Most recently, BaTiO₃ (BTO) nonvolatile phase shifters have also been demonstrated with multi-level states with a switching energy as low as 4.6 pJ and excellent controllability[35]. While a promising advancement, they require a reset sequence consisting of 10,000 pulses with a duration totaling hundreds of microseconds before switching states. Phase shifters based on BTO also require about a ~1 mm phase shifter length to achieve a π phase shift, which is challenging to scale and achieve high-speed operation.

In this work, we have demonstrated a nonvolatile III–V-on-silicon memresonator used for programmable photonic memory operating at record low switching energy (0.15–0.36 pJ), sub-nanosecond switching times enabling high-speed, energy-efficient in-memory computing within silicon photonic neural networks. These nonvolatile optoelectronic memory devices save a great deal of energy by reducing the power consumption involved in programming phase shifters within photonic integrated circuits. By using short voltage pulses to permanently switch the state of this device, the energy typically lost in idle power consumption is saved throughout the duration of an inference task. For example, after a write pulse is applied to the memresonator, the device will retain its state until another voltage pulse is used to write a different weight value. In this way, it is worth reiterating that no idle power consumption is wasted in between reading or writing the weight value stored within the memresonator.

In addition, these nonvolatile photonic phase shifters can act as weights within silicon photonic neural networks which can be updated in real-time, enabling algorithms such as error backpropagation to be executed directly on-chip, greatly optimizing the acceleration of silicon photonic neural networks. Another significant distinction is that the memory is directly on the same chip as the phase shifter, enabling the capability to perform in-memory photonic computing. This avoids the optical-to-electrical conversion losses involved in going to off-chip memory in between each data set used to train a neural network. For instance, in the gradient descent algorithm, predicted values are subtracted from the actual values of the neural network in between training iterations to calculate the cost function. This device can save a substantial amount of energy involved in fetching data typically stored in an external memory chip like static random-access memory (SRAM) or dynamic random-access memory (DRAM) to calculate the cost function in between training iterations. Another area in machine-

learning this device may apply to is transfer learning, described as the practice of re-using a pre-trained neural network instead of training one from scratch to reduce latency and save computational resources[50,51]. Given that the weights in the backbone layer are fixed, they would benefit from being stored in on-chip memory such as with these memresonators. Also, these memresonators can simultaneously be used for the trainable portion of the neural network since they are also capable of being switched at high speeds and with low energy.

Lastly, these devices were developed on a heterogeneous III–V-on-silicon platform, which allows for the co-integration of non-linear active optoelectronic devices, such as lasers and modulators, directly on the same chip as a silicon photonic neural network or an optical FPGA[52]. Since these types of photonic integrated circuits do not inherently need to transmit optical signals off-chip, we gain a significant advantage by integrating the light source directly on-chip. This technology can immensely improve the energy efficiency, stability, and scalability of integrated photonic processors, advancing their potential for use in next-generation HPCs and edge computing.

Future designs will feature device and structural design improvements in order to reduce switching voltages and improve the extinction ratio. The total voltage applied across the device distributes over the Al₂O₃ layer, semiconductor layers (n-GaAs, p-Si), and metal/semiconductor contact layers. By reducing the active area of the device and thickness of the Al₂O₃ layer, optimizing the semiconductor layers' doping concentrations and thicknesses, and improving the quality of the metal/semiconductor contact interface and the Al₂O₃ layer, we can reduce the switching voltage[53]. Also, by increasing the critical coupling within the microring resonator, we can achieve a better extinction ratio and a smaller FWHM. We are also working on improving the waveguide losses on our platform, which will also improve the Q and extinction ratio. Also, TEM images will be taken to investigate the conductive filament formation within these devices and study the physical processes behind the switching mechanisms in these devices. These studies will aid in the design of future devices such as the selection of the resistive-switching oxide material.

Another design change will be to integrate a field-effect transistor in series with the memristor to be able to apply voltage pulses on the device with control of the device current. Integrating these devices with a MOS field-effect transistor (MOSFET) in a one-transistor one-resistor (1T1R) configuration to reliably control the current flow in the device without external circuitry. Within a 1T1R configuration, a MOSFET is connected in series with a memristor and is used to limit the current in the memristor by applying a gate voltage on the MOSFET to modulate the channel length and allow only a certain amount of current to flow through the MOSFET channel. This will play a significant role in improving the control of switching by controlling the amount of current able to conduct in the device. It will also enable multiple intermediate resistance states by using different voltage pulse parameters to select different states while the MOSFET protects the device from permanent breakdown with different voltage pulses. Lastly, memristors can also be integrated within Mach–Zehnder interferometers as an alternative form of a nonvolatile phase shifter also

commonly used within silicon photonic neural networks, quantum computing circuits, and FPGAs[53].

## Methods

Microring resonators with a diameter of $20\,\mu m$ were measured on a copper stage with III–V side up. The experimental set-ups for the measurements taken are shown in Supplementary Fig. 1. Electrical measurements were taken with an Agilent B1500A semiconductor device analyzer, including a B1525A HV-SPGU high-voltage pulse generator. GSG RF probes (Cascade Microtech ACP–40) were used to probe the devices and measure the high-speed response. Optical power measurements were taken using a Newport 2936-R optical power meter. The device was designed with input and output grating couplers, which had about 6 dB of loss each at peak transmission. A Santec TSL-510 tunable laser is used to illuminate the input grating coupler with a cleaved fiber. The laser wavelength is swept and the output of the device is measured through the output grating coupler which is coupled to an optical power meter.

To measure the switching speed of the memresonator, we couple light coming from a tunable laser at the resonant wavelength of the memresonator into the input grating coupler. We then apply voltage pulses from a Keysight M8195A Arbitrary Waveform Generator to read and write the memristor, and couple light coming from the output grating coupler into a high-speed photodiode which is then connected to a Tektronix 8 GHz real-time oscilloscope. A 100 ns wide, 2 V amplitude pulse was used to read the memresonator in the retention time and endurance measurements.

## Data availability

All data are available in the main text or the supplementary materials.

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

## Acknowledgements

The authors want to thank Thomas Van Vaerenbergh, Marco Fiorentino, Sagi Mathai, and Sri Priya Sundararajan for the insightful discussions and input towards this paper. The authors acknowledge that they received no funding in support for this research.

## Author contributions

B.T. conceived the project and D.L. and R.G.B. supervised the project. B.T., D.L. and Z.F. designed the experiments. B.T. performed the experiments and wrote the manuscript. D.L., S.C., Z.F., X.S., J.P.S. and R.G.B. provided helpful suggestions. S.C. and Z.F. provided theoretical analysis. B.T. and Z.F. provided graphics and plots. All authors contributed to the preparation of the manuscript.

## Competing interests

The authors declare no competing interests.
