## [Peer Review File · Nature Communications]

High-Speed and Energy-Efficient Non-Volatile Silicon Photonic Memory Based on Heterogeneously Integrated MemresonatorReviewer #1 (Remarks to the Author):

In this work, the authors present a novel optical memory unit which combines a PN ring modulator with an electronic memristor. The device shows very competitive switching speeds and switching energies and has potential to have a large impact in the field of nonvolatile optical memory and analog optical computing. The manuscript is well written, but there are several points that the authors should address in order to present their work fairly against other optical memory technologies.

Major comments:

- On page 17, line 309-310, the authors state, "The bandwidth of similar III-V/Si MOS-type microring modulators have been measured to be as high as 28 GHz." Is this at all relevant to this work? The fact that the materials used are the same is not in any way related to the switching mechanism. In fact, if the readout is dependent on current injection like in a PIN diode, carrier recombination will limit the bandwidth. Better comparisons would be with electronic memristors comprised of the same materials and using the same switching mechanism or with a forward biased PIN microring modulator.

- The comparison presented in table 1 does not fairly review the various technologies the authors compare their work against:

1. It is unfair to claim that thermo-optic phase shifters have an insertion loss of 6.5dB. This approach is being used in quantum photonic circuits precisely because it is the only type of modulator (perhaps also LNOI) that has very low insertion losses. For example, this thermo-optic work from 2013 has a 0.5dB insertion loss, compact footprint limited by the adiabatic couplers rather than the thermal microheaters, switching speed of 2.4 us, and switching energy of 30.5 nJ (<https://doi.org/10.1364/OL.38.000733>). Other works with larger footprint are much better in terms of insertion loss.

2. How is the footprint being calculated? Since the device discussed is an MRR, shouldn't footprint be defined as μm^2 rather than μm ?

3. PCM devices can show a π phase shift in $<12 \mu\text{m}$ and insertion loss of 0.3dB as shown by Rios et al. (<https://link.springer.com/article/10.1186/s43074-022-00070-4>).

4. The insertion loss claimed for the column "Memresonator (This Work)" is not fair since the wavelength shift from switching the authors demonstrate is smaller than the FWHM of the resonance. Looking at Fig 2e, the insertion loss at the position of highest ER ($\Delta\lambda = 0$) is -4 dB when comparing the HRS vs the LRS. This is the insertion loss that should be used since it represents the insertion loss of the optical signal that is actually being modulated by the MRR.

Other comments:

- Is the transmission of the MRR directly dependent on the read current? It would be helpful to show a plot of how the spectrum of the MRR changes with increasing read current. I see that there is a similar plot in the supplementary, but this does not show the wavelength shift and the quality factor of the ring simultaneously. Also, the authors only present the read current as a function of voltage in the HRS (the best case). They should also show this for the LRS. These plots should be in the main text, not buried in the supplementary since they are important to consider when comparing optical memory technologies.

- Can the ER of the device be improved? The results are very promising in general, but if the change in transmission is limited to between 10% and 27% like what is shown in this work, it is difficult to see a broad application for this technology. The authors should spend more time discussing the challenges and approaches that could improve their current device in terms of ER, insertion loss, multi-levels, etc. and less time in the discussion section promoting the additional features and planned experiments that could be added.

Reviewer #2 (Remarks to the Author):

This paper by Tossoun et al. reports the fabrication and characterization of a photonic ring resonator with an Al₂O₃-based resistive memory. The authors show that the device exhibits memristive behavior with fast (\sim ns) and low-energy (\sim pJ) switching. Coupled with a III/V-on-silicon MOSCAP structure, this leads to a nonvolatile resonance shift in the ring, which can be read out optically. Such a device could find use in many situations where a large number of compact, nonvolatile phase shifters are needed, e.g. for trimming in large low-power photonic circuits, or for photonic neural networks based on a compute-in-memory (CIM) architecture. The use of a standard electronic component (resistive memory) in a proven photonics process (III/V-on-silicon) opens the possibility of future large-scale integration.

I find the concept of this paper very appealing and potentially high-impact, and hope it can be accepted eventually. However, I have a number of serious reservations with the paper in its current form, primarily focused on (1) the quality of presentation, which needs improvement, and (2) a number of incorrect or unexplained technical claims. I would not accept the paper as is, but would be willing to review an appropriately revised manuscript.

(1) Writing.

This paper was tedious to read through and needs significant improvement in writing quality. This is a general comment and it is hard to give concrete advice without rewriting the paper. I would suggest that a colleague review the manuscript writing in detail to improve it. But specific examples I can give would be:

- * Improper passive voice in abstract "introduced for the first time are..."
- * "And their ability to efficiently compute MAC operations" — this sounds weird, and is not obvious what is meant by this being a "limitation"
- * Some paragraphs have strange introductory or concluding sentences, for example, the conclusion to the paragraph and section on Line 97 feels unnatural.
- * The section "Working Mechanism" is really long and wordy, involve equations that are never used in the paper, and overall made the reader (me) want to give up before even getting to the results (consider putting explanatory material after the results, not before).
- * The paragraph break in Lines 292-3 is incredibly abrupt and artificial. It would be more logical to break into one paragraph for SET, the other for RESET. The description of the experiment felt very much like a student lab report, straightforward but also mechanical and tedious to read.

(2) Technical Claims

- * (Line 57) A processing timescale of "femtoseconds" is claimed. This is obviously wrong, since cavity-based devices have timescales limited by the cavity lifetime (\sim picoseconds), and waveguide-based circuits are limited by the speed-of-light transit time (\sim picoseconds). Excitable laser dynamics (ref. 12) should be picosecond.
- * (Line 77) Does RRAM really have 10^{12} cycle endurance? I think that it is usually much less than this.
- * I understand why the LRS and HRS states form, but why does the IRS form? Why is there only one IRS rather than multiple states or a continuum? Is this seen elsewhere in the electronic memristor community?
- * (Line 201) While the carriers are concentrated near the interface, I do not think that their increase with position z is exponential (correct me if I am wrong). Please refrain from using the word "exponential" for rapidly changing functions that are not actually exponential.
- * (Line 217) The Drude-Lorentz model is highly inaccurate for free-carrier dispersion in semiconductors. For silicon, the Soref-Bennett formula is used [<https://doi.org/10.1109/JQE.1987.1073206>], while direct-bandgap materials like GaAs see effects due to both free-carrier dispersion and band-filling [<https://doi.org/10.1109/3.44924>]. In fact, the MOSCAP modulation efficiency increases greatly due to band-filling when the bandgap is closer to the carrier wavelength, e.g. in a material like InGaAsP.
- * (Line 243) What is Gamma?
- * (Line 264) What is the voltage for this L_{π} ? In general, $V_{\pi}L$ seems like the better figure of

merit to quote here.

* (Line 280-292) How do you calculate the switching energy and readout energy? Why do you need a voltage pulse when reading the cavity? Why is the read voltage pulsed rather than always "on"? Does keeping it on degrade the memristor?

* (Line 308) Why is this type of memristor orders-of-magnitude faster than other memristors, as mentioned in this paragraph? Does it have to do with the small oxide dimensions and high thermal diffusion constant?

* (Line 354) The BTO figure here is not consistent with the table.

* (Line 357) It is cheating to compare a ring-based photonic memory to nonresonant devices. Other photonic memories could be enhanced by using a ring, too. Please compare in terms of a fair figure of merit such as L_{π} .

* (Table 1) Why is the retention time for the memresonator so low? Is this just the time it took you to exhaust the device with 10^4 switching events (Fig. 4)? How long does the memresonator hold its state if you are not constantly switching it? I would expect much longer.

* (Line 380) This justification makes no sense. In compute-in-memory, activations are not stored in ring resonators, the weights are. So why does an optical memristor save energy when fetching the intermediate activations?

* (Line 383) Transfer learning is not a "type of neural network". Reference on transfer learning?

Reviewer #1 (Remarks to the Author):

In this work, the authors present a novel optical memory unit which combines a PN ring modulator with an electronic memristor. The device shows very competitive switching speeds and switching energies and has potential to have a large impact in the field of nonvolatile optical memory and analog optical computing. The manuscript is well written, but there are several points that the authors should address in order to present their work fairly against other optical memory technologies.

Thank you for your comments and sharing your valuable insights with us. We have addressed your suggestions and made changes to try and make more fair comparisons with other optical memory technologies. Please find our replies to your comments as well as the red-lined changes in our manuscript below.

Major comments:

- On page 17, line 309-310, the authors state, "The bandwidth of similar III-V/Si MOS-type microring modulators have been measured to be as high as 28 GHz." Is this at all relevant to this work? The fact that the materials used are the same is not in any way related to the switching mechanism. In fact, if the readout is dependent on current injection like in a PIN diode, carrier recombination will limit the bandwidth. Better comparisons would be with electronic memristors comprised of the same materials and using the same switching mechanism or with a forward biased PIN microring modulator.

Thank you for pointing this out. It is true that the switching mechanism described in this manuscript is different and cannot be compared directly with the bandwidth of the devices mentioned in this statement. We have removed this statement to avoid confusion.

In the case of the switching speed in the non-volatile operation, memristors are typically limited by (1) the atomistic processes within the memristor stack, (2) the time constant associated with Joule heating causing the conductive filament to rupture, and (3) parasitic capacitances. More detail on these processes and their fundamental limits are discussed in the following reference:

Menzel, Stephan, et al. "The ultimate switching speed limit of redox-based resistive switching devices." *Faraday Discussions* 213 (2019): 197-213.

We have included this information in the manuscript. We have also added comparisons of electronic memristors, however, there are no other electronic memristors that we are aware of made from the same materials. The closest device we found is a Si/SiO₂/Si based memristor, which had a SET speed of 7.6 μ s and a RESET speed of 490 μ s.

Li, Can, and Qiangfei Xia. "Three-dimensional crossbar arrays of self-rectifying Si/SiO₂/Si memristors." *Handbook of memristor networks* (2019): 791-813.

- The comparison presented in table 1 does not fairly review the various technologies the authors compare their work against:

1. It is unfair to claim that thermo-optic phase shifters have an insertion loss of 6.5dB. This approach is being used in quantum photonic circuits precisely because it is the only type of modulator (perhaps also LNOI) that has very low insertion losses. For example, this thermo-optic work from 2013 has a 0.5dB insertion loss, compact footprint limited by the adiabatic couplers rather than the thermal microheaters, switching speed of 2.4 us, and switching energy of 30.5 nJ (<https://doi.org/10.1364/OL.38.000733>). Other works with larger footprint are much better in terms of insertion loss.

That is a fair point, thanks for informing us of these references. We have updated Table 1 with the update insertion loss numbers.

2. How is the footprint being calculated? Since the device discussed is an MRR, shouldn't footprint be defined as μm^2 rather than μm ?

The footprint was previously defined as the device length. To make this a more clear and fair comparison, we have changed this to L_π , the phase shifter length required to achieve a π phase shift instead.

3. PCM devices can show a π phase shift in $<12 \mu\text{m}$ and insertion loss of 0.3dB as shown by Rios et al. (<https://link.springer.com/article/10.1186/s43074-022-00070-4>).

Thanks for bringing this article to our awareness. We have now updated these numbers in Table 1 and included this data in Lines 287-289.

4. The insertion loss claimed for the column "Memresonator (This Work)" is not fair since the wavelength shift from switching the authors demonstrate is smaller than the FWHM of the resonance. Looking at Fig 2e, the insertion loss at the position of highest ER ($\Delta\lambda = 0$) is -4 dB when comparing the HRS vs the LRS. This is the insertion loss that should be used since it represents the insertion loss of the optical signal that is actually being modulated by the MRR.

It is true that the insertion loss at the highest ER is close to -4 dB. However, we could operate at the wavelength where the insertion loss is lowest ($<-1 \text{ dB}$) at the expense of a lower ER. Given this, we can also improve the ER by improving the waveguide losses and redesigning the devices to better achieve critical coupling. The insertion loss that was

claimed was calculated in Supplementary Note S2, using this reference:

McKinnon, W. R., D-X. Xu, C. Storey, E. Post, A. Densmore, A. Delâge, P. Waldron, J. H. Schmid, and S. Janz. "Extracting coupling and loss coefficients from a ring resonator." *Optics express* 17, no. 21 (2009): 18971-18982.

The results show the insertion losses for devices with different diameters and bias voltages in Fig. S2.

Other comments:

- Is the transmission of the MRR directly dependent on the read current? It would be helpful to show a plot of how the spectrum of the MRR changes with increasing read current. I see that there is a similar plot in the supplementary, but this does not show the wavelength shift and the quality factor of the ring simultaneously. Also, the authors only present the read current as a function of voltage in the HRS (the best case). They should also show this for the LRS. These plots should be in the main text, not buried in the supplementary since they are important to consider when comparing optical memory technologies.

The transmission and wavelength shift of the MRR is dependent on the read current in the IRS and LRS. However, they are not directly dependent on the read current in the HRS. In the HRS, the wavelength shift is dependent on the plasma dispersion effect and carrier accumulation inside of the waveguide as indicated in Formula 4. Carrier accumulation is a field effect, so it is dependent on voltage, not current as indicated in Formulas 1-3. We have included a plot which demonstrates the MRR spectrum as a function of read voltage applied in the HRS and the LRS in the main text in Figure 4 (a) and (d).

- Can the ER of the device be improved? The results are very promising in general, but if the change in transmission is limited to between 10% and 27% like what is shown in this work, it is difficult to see a broad application for this technology. The authors should spend more time discussing the challenges and approaches that could improve their current device in terms of ER, insertion loss, multi-levels, etc. and less time in the discussion section promoting the additional features and planned experiments that could be added.

As mentioned in the response in point 4, the ER can be improved in a few different ways: material choice, device design, and most importantly, material quality the ER can be improved by making some changes to the device design. For example, by increasing the critical coupling of the microring resonator, we can achieve a better extinction ratio and a smaller FWHM. The waveguide losses on these wafers were higher than ideal due to fabrication imperfections. We are working on improving our waveguide losses by making

changes to the dry etch recipe used for the ICP, which will also improve the Q and extinction ratio. Also, applying a larger read voltage pulse will cause a stronger wavelength shift and a corresponding larger change in the optical transmitted power. We have included some of these in the comments in the discussion section.

Reviewer #2 (Remarks to the Author):

This paper by Tossoun et al. reports the fabrication and characterization of a photonic ring resonator with an Al₂O₃-based resistive memory. The authors show that the device exhibits memristive behavior with fast (~ns) and low-energy (~pJ) switching. Coupled with a III/V-on-silicon MOSCAP structure, this leads to a nonvolatile resonance shift in the ring, which can be read out optically. Such a device could find use in many situations where a large number of compact, nonvolatile phase shifters are needed, e.g. for trimming in large low-power photonic circuits, or for photonic neural networks based on a compute-in-memory (CiM) architecture. The use of a standard electronic component (resistive memory) in a proven photonics process (III/V-on-silicon) opens the possibility of future large-scale integration.

I find the concept of this paper very appealing and potentially high-impact, and hope it can be accepted eventually. However, I have a number of serious reservations with the paper in its current form, primarily focused on (1) the quality of presentation, which needs improvement, and (2) a number of incorrect or unexplained technical claims. I would not accept the paper as is, but would be willing to review an appropriately revised manuscript.

We thank you for your valuable feedback which has greatly helped us strengthen the quality of this manuscript. We have responded to your suggestions below and have made note of where changes have been made to the manuscript. Please also find the red-lined manuscript below.

(1) Writing.

This paper was tedious to read through and needs significant improvement in writing quality. This is a general comment and it is hard to give concrete advice without rewriting the paper. I would suggest that a colleague review the manuscript writing in detail to improve it. But specific examples I can give would be:

* Improper passive voice in abstract "introduced for the first time are..."

We have changed this sentence to be in the active voice.

* "And their ability to efficiently compute MAC operations" — this sounds weird, and is not obvious what is meant by this being a "limitation"

We have corrected this on lines 46-47 to be more clear.

* Some paragraphs have strange introductory or concluding sentences, for example, the conclusion to the paragraph and section on Line 97 feels unnatural.

We have adjusted these paragraphs to have more natural beginnings and endings.

* The section "Working Mechanism" is really long and wordy, involve equations that are never used in the paper, and overall made the reader (me) want to give up before even getting to the results (consider putting explanatory material after the results, not before).

We have simplified the wordage and moved a large portion of this section into the Methods section.

* The paragraph break in Lines 292-3 is incredibly abrupt and artificial. It would be more logical to break into one paragraph for SET, the other for RESET. The description of the experiment felt very much like a student lab report, straightforward but also mechanical and tedious to read.

We have made some edits to make these paragraphs flow in a way that is easier to read.

(2) Technical Claims

* (Line 57) A processing timescale of "femtoseconds" is claimed. This is obviously wrong, since cavity-based devices have timescales limited by the cavity lifetime (~picoseconds), and waveguide-based circuits are limited by the speed-of-light transit time (~picoseconds). Excitable laser dynamics (ref. 12) should be picosecond.

Thank you for the correction. We have changed this in the manuscript.

* (Line 77) Does RRAM really have 10^{12} cycle endurance? I think that it is usually much less than this.

While it may be typical to have a lower cycle endurance, we have claimed the highest endurance measured which was to be 10^{12} as shown in this reference:
Goswami, Sreetosh, et al. "Robust resistive memory devices using solution-processable metal-coordinated azo aromatics." Nature materials 16.12 (2017): 1216-1224.

Though, it is a good point that this is an outlier and not what is typical. More typically, resistive switching (RS) devices have exhibited 10^6 - 10^8 cycles quite reliably as noted in this review paper on RS devices:

Lanza, Mario, et al. "Standards for the characterization of endurance in resistive switching devices." *ACS nano* 15.11 (2021): 17214-17231.

We have updated this number in the manuscript in line 78 and added this reference.

* I understand why the LRS and HRS states form, but why does the IRS form? Why is there only one IRS rather than multiple states or a continuum? Is this seen elsewhere in the electronic memristor community?

The IRS forms because by applying a current compliance, the size or volume of the conductive filaments can be controlled. This mechanism is well-documented in the literature and a description can be found in the following reference:

Yang, Yuchao, et al. "Observation of conducting filament growth in nanoscale resistive memories." *Nature communications* 3.1 (2012): 732.

The reference mentions within it: "These arguments also predict that it is possible to control the filament growth by adjusting the programming current. As shown in Fig. 3, four different devices with similar structures were programmed with different programming currents (Fig. 3a–d), and an increase of the programming current indeed resulted in an increase in filament size, both in length and in width (Fig. 3e–h)." We have added this reference to line 174.

* (Line 201) While the carriers are concentrated near the interface, I do not think that their increase with position z is exponential (correct me if I am wrong). Please refrain from using the word "exponential" for rapidly changing functions that are not actually exponential.

It is correct that the carriers that are concentrated near the interface are not necessarily exponential. Thanks for notifying us of this, and we have made the correction in line 373.

* (Line 217) The Drude-Lorentz model is highly inaccurate for free-carrier dispersion in semiconductors. For silicon, the Soref-Bennett formula is used [<https://doi.org/10.1109/JQE.1987.1073206>], while direct-bandgap materials like GaAs see effects due to both free-carrier dispersion and band-filling [<https://doi.org/10.1109/3.44924>]. In fact, the MOSCAP modulation efficiency increases greatly due to band-filling when the bandgap is closer to the carrier wavelength, e.g. in a material like InGaAsP.

It is correct that free-carrier dispersion and band-filling both play a role in III-V semiconductor materials. With that said, we have added Formulas 3 and 4 to include the band-filling effect in the GaAs material. We also modified Formula 5 to include the independent effective index of refraction changes in the Si and the GaAs.

* (Line 243) What is Gamma?

Gamma is the confinement factor of the fundamental mode within the waveguide. We have added this to line 434 after Formula 6.

* (Line 264) What is the voltage for this L_{π} ? In general, $V_{\pi}L$ seems like the better figure of merit to quote here.

The read voltage used was 2 V as described in Supplementary Note S5. Therefore, the $V_{\pi}L$ is $2 V_{\pi} * 0.35 \text{ mm} = 0.7 V_{\pi}L$. We have changed to $V_{\pi}L$ in the manuscript on line 202.

* (Line 280-292) How do you calculate the switching energy and readout energy? Why do you need a voltage pulse when reading the cavity? Why is the read voltage pulsed rather than always "on"? Does keeping it on degrade the memristor?

The calculations for the switching and readout energy can be found in Supplementary Note S1. A voltage pulse is needed to create a change in the current flow within the memristor. This current flow then determines the degree of the phase shift. The readout voltage is pulsed to save static power consumption. Theoretically, we only need to apply a read voltage on the phase shifters during computations.

* (Line 308) Why is this type of memristor orders-of-magnitude faster than other memristors, as mentioned in this paragraph? Does it have to do with the small oxide dimensions and high thermal diffusion constant?

Actually, the line does not claim that this device is order-of-magnitude faster than other memristors. It reads: "The measured switching speed of these devices is over two orders of magnitude faster than the fastest non-volatile photonic phase shifters and is comparable to all-electronic memristor devices made of similar materials." It mentions that the switching speed is over two orders of magnitude faster than the fastest non-volatile *photonic* phase shifters (based on PCM), but not other electronic memristors. It goes on to say the switching speed is comparable to other all-electronic memristors. The main reason that this device is faster than devices based on PCM is due to the fact that PCM requires a high current and needs to be heated long enough to change the phase from amorphous to crystalline as mentioned in lines 290-291, which is a much slower effect (typically $> 100 \text{ ns}$) than the switching speed of resistive switching based memristors ($< 1 \text{ ns}$).

* (Line 354) The BTO figure here is not consistent with the table.

The figure for BTO under Table 1 for Device Footprint describes the physical length of the phase shifter. The figure in line 354 refers to the effective length of the phase shifter to achieve a π phase shift, or L_{π} . In order to make this clearer, we have changed the row labeled "Device Footprint" to reflect L_{π} .

* (Line 357) It is cheating to compare a ring-based photonic memory to nonresonant devices. Other photonic memories could be enhanced by using a ring, too. Please compare in terms of a fair figure of merit such as L_{π} .

It is true that comparing microrings and MZIs are not so straightforward. However, these BTO devices were actually also made on racetrack resonators with a 50 μm bend radius. Please find in the reference here: "(Geler-Kremer, Jacqueline, et al. "A ferroelectric multilevel non-volatile photonic phase shifter." Nature Photonics 16.7 (2022): 491-497)"

We have also noted the L_{π} of both the BTO in Line 295 and the memresonator in line 202. To be clearer, we have reiterated the L_{π} of the memresonator in the same line and made a direct comparison between the two. As mentioned in the last comment, to also make comparisons clearer and fairer, we have changed the "Device Footprint" metric in Table 1 to L_{π} , the phase shifter length required to achieve a π phase shift instead.

* (Table 1) Why is the retention time for the memresonator so low? Is this just the time it took you to exhaust the device with 10^4 switching events (Fig. 4)? How long does the memresonator hold its state if you are not constantly switching it? I would expect much longer.

The device holds its states up to 12 hours when it is not being switched, as demonstrated in Figure 4(a). Given the state of the art in photonic phase shifters, this is a reasonably good result considering it is the first generation of these types of phase shifters. For instance, the non-volatile photonic phase shifter made with BTO was recently published in Nature Photonics (Geler-Kremer, Jacqueline, et al. "A ferroelectric multilevel non-volatile photonic phase shifter." Nature Photonics 16.7 (2022): 491-497) and showed a retention time of 10 hours. Furthermore, plenty of electronic memristors are published with similar retention times. Here are a few papers for examples:

A large part of this is due to the experimental set up that was used. Having a more mechanically stable set up as well as thermal stability would allow us to further increase the retention time measurement. This was also mentioned in lines 208-209.

* (Line 380) This justification makes no sense. In compute-in-memory, activations are not stored in ring resonators, the weights are. So why does an optical memristor save energy when fetching the intermediate activations?

It is correct that intermediate activations are not stored while training. What is meant by this line is that the device stores the output results of the final layer of the neural network while the network is being trained. Rather than fetching the stored results from an external memory chip to calculate the cost function, the results can be stored directly on-chip after a training iteration is finished. Then, the gradient of the cost function can be calculated by

subtracting the new results with the stored results on-chip after updating the weights in each training step. This saves us the energy of needing to go to the off-chip external memory between each training iteration. This has been clarified in the manuscript in lines 320-325

* (Line 383) Transfer learning is not a "type of neural network". Reference on transfer learning?

This is correct. Transfer learning is not a type of neural network, but rather a technique used in machine learning to improve the performance of a related task. We have modified this and added a reference on transfer learning to the paper.

Reviewer #1 (Remarks to the Author):

The authors have address the majority of my concerns. However, they still do not present a fair comparison in their table for insertion loss. They refuse to report their experimentally measured insertion loss, which is 4 dB, and instead are extracting it from the intrinsic Q of the ring which ignores the coupling losses. This in itself is unfair since the coupling losses are a necessary part of the memresonator for it to function(i.e., there is no modulation if you don't couple light to the ring). Therefore, they are giving themselves an unfair advantage when comparing against other technologies in the field which directly report insertion loss.

Additionally, since they are comparing against devices which actually can experimentally achieve a pi phase shift, they have chosen to report "Lpi" which is fine. However, they are also not scaling their insertion loss to match this Lpi they report in their table for the memresonator. To correct this, their insertion loss should be (at minimum) the following:

$$350\mu\text{m} * 0.048\text{db} / (2*\pi*10\mu\text{m}) = 0.27 \text{ dB insertion loss}$$

However, the true insertion loss of their device is 4dB since this is their chosen operating point throughout the paper to achieve maximum modulation.

I make a big deal about having an accurate comparison since future review and research articles will simply copy whatever values are reported in Table 1 without actually reading this article and understanding these details.

Reviewer #1 (Remarks to the Author):

The authors have address the majority of my concerns. However, they still do not present a fair comparison in their table for insertion loss. They refuse to report their experimentally measured insertion loss, which is 4 dB, and instead are extracting it from the intrinsic Q of the ring which ignores the coupling losses. This in itself is unfair since the coupling losses are a necessary part of the memresonator for it to function(i.e., there is no modulation if you don't couple light to the ring). Therefore, they are giving themselves an unfair advantage when comparing against other technologies in the field which directly report insertion loss.

Additionally, since they are comparing against devices which actually can experimentally achieve a pi phase shift, they have chosen to report "Lpi" which is fine. However, they are also not scaling their insertion loss to match this Lpi they report in their table for the memresonator. To correct this, their insertion loss should be (at minimum) the following:

$$350\mu\text{m} * 0.048\text{db} / (2*\pi*10\mu\text{m}) = 0.27 \text{ dB insertion loss}$$

However, the true insertion loss of their device is 4dB since this is their chosen operating point throughout the paper to achieve maximum modulation.

I make a big deal about having an accurate comparison since future review and research articles will simply copy whatever values are reported in Table 1 without actually reading this article and understanding these details.

Thank you again for sharing your concern with us. We have taken your suggestions and made corrections to try and make a fair comparison on insertion loss with other optical memory technologies in Table 1. Please find our reply to your comments as well as the red-lined changes in our manuscript below.

We have corrected the number reported for insertion loss in Table 1 from 0.048 dB to 0.27 dB, as suggested. We have also included the following footnote: "***The total insertion loss is measured to be 4 dB at the operating point of interest when including coupling losses."